# OpenReview forum: "AICrypto: A Comprehensive Benchmark for Evaluating Cryptography Capabilities of Large Language Models"
_ICLR.cc/2026/Conference — Submitted to ICLR 2026_

### Official Review · Reviewer_w8pB · 2025-10-26

**Soundness:** 3
**Presentation:** 2
**Contribution:** 3
**Rating:** 4
**Confidence:** 4

**Summary:**

This paper introduces AICrypto, a cryptography‑focused benchmark intended to comprehensively evaluate LLM capabilities across (i) 135 MCQs, (ii) 150 CTF challenges spanning nine crypto categories, and (iii) 18 proof problems drawn from three university exams. It also proposes an agentic evaluation harness for the CTFs and reports results for 17 leading LLMs. The key finding is that top models match or beat humans on MCQs, approach human performance on routine proofs, but lag substantially on CTFs that demand dynamic reasoning and program analysis.

**Strengths:**

- MCQs probe factual knowledge; proofs test formal reasoning; and CTFs stress real‑world exploitation skills.
- Mitigating Data Contamination. The paper documents rewriting/verification for MCQs and expert authorship for proofs, with manual quality checks, to avoid data contamination of the benchmark.

**Weaknesses:**

1. Helper scripts change the original CTF question. For many CTFs, the benchmark injects helper.py/.sage, which meaningfully lowers parsing/IO friction relative to human play and could inflate LLM success on certain categories. An ablation “with vs. without helper” would quantify this effect.
2. Proof scoring is manual and single‑pass. While the authors explain why LLM‑as‑grader is unreliable, the paper does not discuss inter‑rater agreement or rubric calibration across graders. Given that subtle logic gaps are a core finding, reporting inter-annotator metrics would bolster credibility. Moreover, manual grading makes running on the Proof subsection not scalable.
3. Limited # of Questions. Each subsection contains only a limited number of questions per sub-category. For example, DLP (under CTF) just contains 10 questions, while there are just 18 Proof Questions (all sub-categories combined), making the results not significant. Reporting the significance of the results or adding more questions would greatly improve the reliability of the benchmark.
4. Saturation on MCQ subset. All LLMs perform exceptionally well on the MCQ subset, with 11 LLMs either outperforming or matching human performance.

**Questions:**

Minor clarity/typo issues:
- “provider insights” in the conclusion, line 485
- “consistently general models” line 318
- Fig 4 instead of Fig 5 (line 196)
- Fig. 11 caption text is incorrect

See weaknesses above

---

> ### Author Response · Authors · 2025-11-23
> **Response to Reviewer w8pB**
>
> We really appreciate your time and insightful comments! We hope the following experiments and discussions can address your questions and concerns.
>
> **Weakness 1: Helper scripts ablation.**
>
> Thank you very much for raising this point. First, the helper scripts are necessary in our setting. Many cryptographic CTF challenges produce very large outputs, such as huge primes or complicated polynomials. The length of this raw output even often exceeds the context window of some LLMs. To keep the framework usable and to control cost, our framework truncates very long command outputs when the LLM runs a command. As a result, for some CTF problems the LLM cannot obtain the complete original output. The helper script therefore becomes necessary for our framework, otherwise we will force the model to handle an incomplete raw output.
>
> Second, in cryptographic CTFs the data loading and basic I/O steps are usually very simple for human players. In most cases, a human only needs to copy and paste the relevant values, and this part is not the focus of the challenge. The core difficulty lies in identifying and exploiting cryptographic vulnerabilities.
>
> **Weakness 2: Proof scoring is manual and single‑pass.**
>
> Thank you for your insightful comment. To address the concern that the proof problems cannot be evaluated automatically and do not have rigorous scoring criteria, we establish rigorous scoring criteria and use the two most powerful LLMs for automated scoring. We then validate the scores against those of human experts and obtain a very high correlation coefficient. To address the single-pass issue, we design a peer-review process for the human expert scoring phase, have the grading model score multiple times, and evaluate each model through multiple experiments. Please refer to the **General Response** for details.
>
> **Weakness 3: Limited # of Questions.**
>
> Thank you very much for pointing this out. Regarding the first concern, our CTF challenges are collected from professional CTF competitions. The distribution of our challenges therefore reflects the distribution in real competitions. The DLP category appears relatively infrequently in actual events. AICrypto is the first comprehensive benchmark that evaluates LLM cryptography capabilities, and the field is still at an early stage, we plan to leave the expansion of challenges in this specialized area for future work.
>
> Regarding your second concern, we have added 12 additional proof problems, bringing the total number of proof problems to 30, please refer to General Response for details. We would like to clarify again that proof problems and CTF challenges are fundamentally different from multiple choice questions; they contain much richer information. In our case study, we provide additional analysis specifically for these problems based on careful manual inspection by human experts.
>
> **Weakness 4: Saturation on MCQ subset.**
>
> Thank you very much for this observation. Although the strongest models already reach or exceed human performance on the MCQs, these results remain informative. Our MCQs test factual knowledge of fundamental cryptographic principles and concepts. The strong performance indicates that current LLMs already master these basics, which provides a useful point for future research and real-world applications.
> In addition, while our main experiments focus on frontier models, the MCQ subset in AICrypto continues to be challenging and valuable for smaller models. We further evaluate Llama-2-7b-Chat and Llama-3-8B-Instruct. Llama-3-8B-Instruct achieves an accuracy of only 32.59% on the MCQs. This suggests that the MCQ subset continues to be a meaningful tool for assessing basic cryptographic knowledge in smaller models, even if frontier models already perform very well.
> | Model                 | Accuracy    |
> |-----------------------|--------|
> | Llama-2-7b-Chat       | 8.89%  |
> | Llama-3-8B-Instruct   | 32.59% |
>
> **Questions:**
> Thank you very much for pointing out these typos. We have corrected them in the newly uploaded revision.
>
> Thank you again for your thoughtful review. Please let us know if you have any additional questions or concerns.

---

### Official Review · Reviewer_1oun · 2025-10-28

**Soundness:** 3
**Presentation:** 3
**Contribution:** 3
**Rating:** 6
**Confidence:** 3

**Summary:**

The paper introduces AICrypto, a comprehensive benchmark for evaluating LLMs' cryptographic capabilities across three task types: 135 multiple-choice questions testing factual knowledge, 150 CTF challenges requiring practical exploitation skills, and 18 proof problems assessing formal reasoning. The authors evaluate 17 state-of-the-art LLMs using an agent-based framework for CTF challenges and expert grading for proofs. Results show that leading models match or exceed human experts on conceptual knowledge and routine proofs but significantly underperform on practical CTF challenges requiring multi-step reasoning and dynamic analysis.

**Strengths:**

1. Comprehensive benchmark design: Three complementary task types provide holistic evaluation of cryptographic competence

2. Rigorous curation process: Expert involvement ensures task quality and prevents data contamination

3. Strong empirical evaluation: 17 models evaluated with human expert baselines for comparison

4. Practical insights: Clear identification of model limitations in numerical computation and multi-step reasoning

5. Reproducibility: Code and dataset made publicly available

**Weaknesses:**

1. Agent framework limitations: The simple agent framework may handicap model performance on CTFs. More sophisticated planning or tool-use strategies aren't explored.

2. Scalability issues: Manual proof grading is unsustainable. The attempted automated grading failure needs addressing for practical benchmark use.

3. Missing analysis: No investigation of whether specific model architectures or training approaches correlate with better cryptographic reasoning.

**Questions:**

1. How sensitive are CTF results to the agent framework choice? Have you tested more sophisticated approaches like ReAct or tree-search based planning?

2. Could formal verification tools (Coq, Lean) be integrated to enable automated proof checking rather than manual grading?

3. The paper mentions some models timing out on polynomial GCD computations (e.g., Figure 18). Could you quantify computational complexity thresholds where models consistently fail?

---

> ### Author Response · Authors · 2025-11-23
> **Response to Reviewer 1oun**
>
> We really appreciate your time and insightful comments! We hope the following experiments and discussions can address your questions and concerns.
>
> **Weakness 1: Agent framework limitations.**
>
> Thank you for raising this point. We understand the concern and have discussed it it in our Limitations section. Our work focuses on examining the cryptographic capabilities of LLMs. Exploring more advanced agent frameworks, including richer planning or tool-use strategies, goes beyond the scope of the current study. We consider this an important direction for future work.
>
> **Weakness 2:  Scalability issues: Manual proof grading is unsustainable.**
>
> Thank you for your comment. We have addressed the concern about full automation of evaluation. We establish rigorous scoring criteria and use the two most powerful LLMs for automated scoring. We then validate the scores against those of human experts and obtain a very high correlation coefficient. Please refer to the **General Response** for details.
>
> **Weakness 3:  Missing analysis.**
>
> Thank you for this insightful comment. Our study evaluates state-of-the-art LLMs such as OpenAI o3 and Google gemini-2.5-pro. These models are commercial and closed source, and their architectures and training procedures are not publicly available. This limitation prevents us from conducting a meaningful analysis of how specific design or training choices correlate with cryptographic reasoning. We agree that this is an important direction. Future work can investigate open-source or self-trained models on our dataset to enable a more detailed analysis.
>
> **Question 1: How sensitive are CTF results to the agent framework choice?**
>
> Thank you for this thoughtful question. As noted in our response to Weakness 1, the primary goal of this work is to evaluate the cryptographic abilities of LLMs. An in-depth exploration of alternative agent frameworks, including approaches such as ReAct or tree-search based planning, falls outside the scope of the current study. This remains a valuable direction for future work.
>
> **Question 2: Could formal verification tools (Coq, Lean) be integrated to enable automated proof?**
>
> Thank you for this thoughtful question. There are several practical obstacles. Taking Lean as an example, Lean, especially Lean 4 with its mathlib library, relies on a specific formal system based on dependent type theory. Only mathematical objects and reasoning that fit within this system can be fully formalized. Many cryptographic concepts do not yet have the necessary foundational components in Lean or mathlib. They may be theoretically expressible, but the required definitions and lemmas are not available, which makes the actual formalization either infeasible or extremely labor intensive. For these reasons, we do not integrate formal verification tools in our evaluation.
>
> **Question 3: Computational complexity thresholds.**
>
> Thank you for the question. In our agent setting, each command execution has a predefined timeout. This timeout is determined from the running time of a correct reference solution. As described in the Appendix, when the correct solution finishes within 60 seconds, we set the timeout to 300 seconds. When the correct solution takes more than 60 seconds, we set the timeout to five times that duration. If an LLM executes a command longer than this threshold, it receives a timeout feedback.
>
> We adopt this design because, in our experiments, models often attempt the brute force methods when they fail to solve a challenge through analysis, which can require unbounded computation. CTF challenges are generally intended to be solved through reasoning about vulnerabilities rather than brute force attacks.
>
> The polynomial GCD examples raised by the reviewer arise only when the model fails to implement the polynomial GCD procedure correctly and instead falls back to querying external factor databases or attempting a brute force search. Because the moduli in the CTF context are extremely large, these fallback behaviors either fail outright or require infeasible time, even though the correct algorithm runs in essentially constant time on these instances. Consequently, while we can guarantee that at least one correct solution exists that finishes within the time limit, we cannot predict the running time of the incorrect strategies chosen by the models. Thus, the observed failures do not reflect any intrinsic computational complexity threshold, but rather the models' selections of impractical brute force approaches.
>
> Thank you again for your thoughtful review. Please let us know if you have any additional questions or concerns.

---

### Official Review · Reviewer_E18C · 2025-10-30

**Soundness:** 2
**Presentation:** 3
**Contribution:** 2
**Rating:** 4
**Confidence:** 3

**Summary:**

This paper presents AICrypto, a benchmark focusing on evaluating large language models (LLMs) on cryptographic tasks. It consists of three types of problems, namely, multiple-choice questions, capture-the-flag, and proof problems. An evaluation on 18 LLMs suggests that these LLMs perform well on multiple-choice questions but still struggle on the other two challenging tasks.

**Strengths:**

- The paper is well-written and easy to follow.

- The presented benchmark covers a range of different cryptographic tasks.

- The evaluation covers a decent amount of recent LLMs, including Gemini 2.5 Pro and Claude Sonnet 4.

**Weaknesses:**

- The dataset size is very small: The dataset contains 35 MCQs, 150 CTFs, and 18 proofs, and then in total fewer than 200 questions. It is unclear whether any conclusions drawn from 200 examples are statistically significant, especially given that there are lots of factors in LLM evaluation (model, prompt, etc) that can easily affect the performance.

- Usefulness is limited due to human-in-the-loop evaluation: MCQs are fully automated but most models have already achieved superhuman performance. The proof problems are interesting and seem to be useful for measuring progress in AI cryptography. However, the evaluation currently requires human supervision, which is extremely expensive.

- Data coverage seems to be poor: Almost all questions are collected from university exams or public cryptography challenges. While valuable, these questions often do not cover many real-world scenarios and are biased towards academic topics.

**Questions:**

See the weakness points above.

---

> ### Author Response · Authors · 2025-11-23
> **Response to Reviewer E18C**
>
> We really appreciate your time and insightful comments! We hope the following experiments and discussions can address your questions and concerns.
>
> **Weakness 1 & 3:  Datasize and coverage**
>
> Thank you for the comment. We would first like to clarify a factual error from the reviewer. **AICrypto contains 135 multiple choice questions rather than 35**. The dataset therefore includes 135 MCQs, 150 CTF challenges, and 18 proofs, for a total of 303 examples.  We also add 12 new proof problems and automatic evaluation (details included in our general response), so AICrypto now contains 30 proofs in total, which further enriches the benchmark.
>
> In addition, our dataset is relatively large compared with contemporary work. For example, Cybench [1] contains 40 CTF challenges. NYU CTF Bench [2] includes 200 challenges, but only 62 of them are related to cryptography. Our dataset already exceeds these by a substantial margin in terms of cryptography focused tasks, and we believe that its size is sufficient to support the conclusions in the paper.
>
> We agree that cryptography involves both practical and theoretical aspects, and that real world scenarios are important. Our benchmark is the first comprehensive effort to evaluate cryptographic capabilities of LLMs, and research in this direction is still at an early stage. Within this scope, we include CTF challenges that simulate practical situations. Expanding the benchmark to include a broader range of realistic tasks is an open problem and a natural direction for future work.
>
> [1] Zhang, Andy K., et al.  Cybench: A Framework for Evaluating Cybersecurity Capabilities and Risks of Language Models. The Thirteenth International Conference on Learning Representations (ICLR 2025).  https://openreview.net/forum?id=tc90LV0yRL
> [2] Shao, Minghao, et al. "Nyu ctf bench: A scalable open-source benchmark dataset for evaluating llms in offensive security." Advances in Neural Information Processing Systems 37 (2024): 57472-57498.
>
> **Weakness 2: Usefulness is limited due to human-in-the-loop evaluation.**
>
> Thank you for the comment. We have addressed the concern about full automation of evaluation. We establish rigorous scoring criteria and use the two most powerful LLMs for automated scoring. We then validate the scores against those of human experts and obtain a very high correlation coefficient. Please refer to the **General Response** for details.
>
> Second, our main experiments focus on frontier models, but the MCQ part of AICrypto remains challenging and useful for smaller models. We further evaluate Llama-2-7b-Chat and Llama-3-8B-Instruct. Llama-3-8B-Instruct achieves an accuracy of only 32.59% on the MCQs. This suggests that the MCQ subset continues to be a meaningful tool for assessing basic cryptographic knowledge in smaller models, even if frontier models already perform very well.
>
> | Model                 | Accuracy    |
> |-----------------------|--------|
> | Llama-2-7b-Chat       | 8.89%  |
> | Llama-3-8B-Instruct   | 32.59% |
>
> Although the strongest models already reach or exceed human level on the MCQs, these results are still informative. Our MCQs measure factual knowledge of fundamental cryptographic principles and concepts. The strong performance shows that current LLMs already master these basics, which provides a useful reference point for future research or their applications.
>
> Third, our CTF challenge is fully automated. State of the art models still leave substantial room for improvement on this CTF benchmark. The best model achieves only around 55% success under a pass@3 evaluation. To our knowledge, this is currently the largest CTF benchmark specifically targeting cryptography.
>
> Thank you again for your thoughtful review. Please let us know if you have any additional questions or concerns.

---

### Official Review · Reviewer_jvuN · 2025-11-01

**Soundness:** 1
**Presentation:** 2
**Contribution:** 2
**Rating:** 2
**Confidence:** 4

**Summary:**

This paper presents a benchmark for evaluating LLMs on cryptography-related tasks, spanning multiple-choice questions on cryptographic concepts, CTF challenges, and proof generation. For CTF tasks, the authors report pass@k and employ an agentic workflow. For proof generation, human experts assess model outputs. The results highlight concrete limitations of current LLMs, including weak mathematical understanding.

**Strengths:**

- The data construction pipeline is solid.
- The experimental evaluation is comprehensive and rigorous.
- The agentic workflow proves effective for CTF tasks.

**Weaknesses:**

1. Data contamination safeguards are time-sensitive. To mitigate contamination, the authors include only challenges and exams from 2023 onward (line 156). However, given we are nearing 2026, this approach is unlikely to ensure contamination-free evaluation. These anti-contamination measures are highly time-sensitive and fragile.
2. Too few test samples per task type. Several categories have very limited instances (e.g., Proof Problems/Signatures contains only one item), which undermines the reliability and statistical significance of the conclusions.
3. A subset of failure modes stems from incorrect mathematical computation. Consider enabling a callable calculator tool to isolate reasoning errors from arithmetic mistakes.
4. What are the differences compared with CryptoBench?

**Questions:**

Line 196: It should reference Figure 5, not Figure 4.

---

> ### Author Response · Authors · 2025-11-23
> **Response to Reviewer jvuN [1/2]**
>
> We really appreciate your time and insightful comments!  We hope the following experiments and discussions can address your questions and concerns.
>
> **Weakness 1: Data contamination safeguards are time-sensitive.**
>
> Thank you for raising this important point. Below, we will show that the presence of official solutions does not lead to higher model success rates in our evaluation. Since our problems are collected primarily from official competitions, the main source of potential data leakage is the release of official solutions. Community blogs also sometimes publish write-ups, but these are often inconsistent in quality and frequently contain errors or incomplete reasoning.
>
> We analyze all CTF challenges that have official online solutions. There are 85 such challenges, which account for 56.67% of the total set (85 out of 150). We also compare model accuracy between problems with official solutions (w OS) and problems without official solutions (w/o OS):
>
> | Model                       | Success Rate w OS  | Success Rate w/o OS |
> |-----------------------------|--------------------|---------------------|
> | o3-high                     | 47.06% (40/85)     | 63.08% (41/65)      |
> | gemini-2.5-pro-preview      | 45.88% (39/85)     | 67.69% (44/65)      |
> | o3                          | 42.35% (36/85)     | 58.46% (38/65)      |
> | o4-mini-high                | 38.82% (33/85)     | 55.38% (36/65)      |
> | o3-mini                     | 27.06% (23/85)     | 35.38% (23/65)      |
> | claude-4.0-sonnet           | 25.88% (22/85)     | 38.46% (25/65)      |
> | o3-mini-high                | 25.88% (22/85)     | 46.15% (30/65)      |
> | o4-mini                     | 25.88% (22/85)     | 44.62% (29/65)      |
> | deepseek-r1                 | 23.53% (20/85)     | 36.92% (24/65)      |
> | claude-4.0-sonnet-thinking  | 22.35% (19/85)     | 40.00% (26/65)      |
> | doubao-seed-1.6-thinking    | 21.18% (18/85)     | 41.54% (27/65)      |
> | claude-3.7-sonnet           | 21.18% (18/85)     | 38.46% (25/65)      |
> | doubao-seed-1.6             | 20.00% (17/85)     | 33.85% (22/65)      |
> | o1                          | 20.00% (17/85)     | 32.31% (21/65)      |
> | gpt-4.1                     | 20.00% (17/85)     | 29.23% (19/65)      |
> | claude-3.7-sonnet-thinking  | 20.00% (17/85)     | 38.46% (25/65)      |
> | deepseek-v3 | 10.59% (9/85)    | 10.59% (9/85)      |
>
> The table indicates that the presence of official solutions does not lead to higher model success rates. In fact, performance is lower on problems with official write-ups. This suggests that data contamination has not introduced meaningful inflation in model performance. We will add this analysis into our manuscript if you think it's helpful.
>
> **Weakness 2: Too few test samples per task type.**
>
> Thank you very much for pointing this out. To address this concern, we have now added 18 proof problems and automatic evaluation. Please refer to our general response for details.
>
> **Weakness 3:  A subset of failure modes stems from incorrect mathematical computation.**
>
> Thank you for highlighting this point. These calculation errors appear only in multiple choice questions. Since the computational load in such questions is small, we evaluate the models using pure natural language. The number of arithmetic mistakes is very limited and remains in the single digits (<5%). Your suggestion is correct that access to a calculator can reduce these errors. Also, considering that the current error rate is already low, the impact on the evaluation results should be very small. In CTF settings, an LLM acting as an agent can directly write code to perform computations, and our proof problems also do not require much arithmetic.

---

> ### Author Response · Authors · 2025-11-23
> **Response to Reviewer jvuN [2/2]**
>
> **Weakness 4:  What are the differences compared with CryptoBench?**
>
>  Thank you for bringing this up. Since no citation is provided, we assume that you are referring to the GitHub repository at https://github.com/xxcg322/CryptoBench.
> We carefully examine this benchmark. As stated on its homepage, CryptoBench evaluates LLM capabilities in understanding and applying cryptographic concepts in practical contexts such as blockchain, cryptocurrencies, and smart contracts. Its main focus is therefore on applications that use cryptography, rather than on cryptography itself. Concretely, the two datasets in the repository contain a small portion of tasks labeled as "Cryptography": 8% (59/727) in multi-choice question and 19% (45/230) in question answering task. Also, note that this CryptoBench does not have a corresponding paper.
>
> Our benchmark, AICrypto, targets a different level of cryptographic understanding. We concentrate on foundational cryptography, including core concepts, vulnerability reproduction, and theoretical reasoning. Our task formats are also more diverse. While CryptoBench uses multiple choice (59 labeled as "Cryptography") and question answering (45 labeled as "Cryptography"), our benchmark includes 135 multiple choice questions, 30 proof problems, and  150 CTF challenges.
>
> **Questions**: Thank you very much for pointing this out. We have already corrected this typo.
>
> Thank you again for your thoughtful review. Please let us know if you have any additional questions or concerns.

---

### Author Response · Authors · 2025-11-23
**General Response [1/2]**

We sincerely thank all the reviewers for their thoughtful comments and feedback. In this general response, we show a set of new results to address concerns on dataset size and automatic evaluation for the proof problems.

## Dataset Size
1. **We add 12 additional proof problems with the same level of difficulty to address this limitation.** The dataset now contains 30 proof problems in total.

2. AICrypto is the first benchmark that provides a comprehensive evaluation of the cryptographic capabilities of LLMs. It includes 135 multiple choice questions, 150 CTF challenges, and 30 proof problems now. Since research on LLMs in cryptography is still at an early stage, **our dataset is relatively large compared with existing work.** For example, Cybench [1] contains 40 CTF challenges. NYU CTF Bench [2] includes 200 challenges, but only 62 of them are related to cryptography. Our dataset exceeds these in terms of the number of cryptography related tasks.

In addition, task types differ in difficulty and diagnostic value. A single CTF challenge or proof problem requires much more complex and integrated reasoning than a multiple choice question. For example, for o3-high, a multiple-choice question might only take around 4,000 tokens, while a proof problem would consume around 40,000 tokens and a CTF challenge could consume even more. Additionally, CTF challenges and proof problems allow us to observe and analyze a wider range of model behaviors.

We would also like to remark that  the difficulty of the proof problems we choose are not merely like university exams or homework problems, which may be thought of as far from cryptographic practice. The proof problems are challenging problems, close to research-level questions in cryptography. If LLMs did well in those problems, they may greatly help the reasoning in solving real research-level questions in cryptography. This is why we think it is important to include proof problems in the benchmark. Also, if you are interested, you can examine the difficulty of the proof problems in our anonymous repository.

[1] Zhang, Andy K., et al.  Cybench: A Framework for Evaluating Cybersecurity Capabilities and Risks of Language Models. The Thirteenth International Conference on Learning Representations (ICLR 2025).  https://openreview.net/forum?id=tc90LV0yRL

[2] Shao, Minghao, et al. "Nyu ctf bench: A scalable open-source benchmark dataset for evaluating llms in offensive security." Advances in Neural Information Processing Systems 37 (2024): 57472-57498.

## Automatic Evaluation
To address the concern that the proof problems cannot be evaluated automatically and do not have rigorous scoring criteria, we propose and demonstrate the effectiveness of the following automatic evaluation system.
1. Our human experts design detailed scoring rubrics for each proof problem. Each problem has 5 points. For example, the rubric for Problem 1 whose focus is on encryption is as follows:

```markdown
### Problem 1
- 2 points. Correctly construct the quantity (e.g. $s = \frac{c_1^{e_2}}{c_2^{e_1}} \bmod N$) that can be used to distinguish the two ciphertexts.
- *1 point. Correctly choose two different messages $m_0, m_1$ (such as $m_0 = 1, m_1 = 2$ or $m_0 = 1, m_1 = -1$ or something else) and correctly compute the distinguish quantity when these messages are encrypted under the new RSA-PKE scheme.
- *1 point. Show that the quantity of two different messages are distinct.
- 1 point. Give a conclusion on why these processes can be used to break the Chosen-Plaintext security.
```
2. Based on these rubrics, our human experts rescore all model output for each problem. We adopt a process in which one expert performs the scoring and another independently reviews it. This ensures both correctness and consistency, which strengthens the robustness of the results.

3. We then use the strongest LLM as automated graders. For each answer, the grader model receives four inputs: the original proof problem, the reference solution, the scoring rubric and the answer to be evaluated. Note that all reference solutions are written and reviewed by human experts to ensure correctness. The grader model provides its reasoning and assigns an integer score from zero to five in accordance with the rubric.
4. We obtain scores by combining two strong models, gpt-5.1 and gemini-3-pro-preview. Each model grades the same answer independently three times, so we obtain six scores in total. We take the average of these six scores as the final score.

We also measure the correlation between human scores and LLM scores on the original 18 proof problems. On 306 samples (18 existing problems × 17 models), **the Pearson correlation between human scores and LLM scores is 0.9025 (p < 1e-10), and the Spearman correlation is 0.8973 (p < 1e-10)**. These high correlations show that our automatic grading strategy is effective.

---

> ### Author Response · Authors · 2025-11-23
> **General Response [2/2]**
>
> We also explore several strategies for aggregating the six LLM scores for each answer, and we compare their correlations with human scores. The strategies are:
>
> 1. **avg** : average of all six scores.
>
> 2. **trimmed1** : remove one highest and one lowest score, then average the remaining four.
>
> 3. **trimmed2** : remove two highest and two lowest scores, then average the remaining two.
>
> 4. **vote** : use the score that appears most frequently as the final score.
>
> 5. **avg_gpt-5.1** :  average of the three scores from gpt-5.1 only.
>
> 6. **avg_gemini-3-pro-preview** : average of the three scores from gemini-3-pro-preview only.
>
> Their correlations with human expert scores are summarized in the following table (all p values are < 1e-10 and therefore omitted):
>
> | Strategy                  | Pearson | Spearman |
> |---------------------------|---------|----------|
> | avg                       | **0.9025**  | **0.8973**   |
> | trimmed1                  | 0.9001  | 0.8965   |
> | trimmed2                  | 0.8947  | 0.8930   |
> | avg_gpt-5.1               | 0.8815  | 0.8857   |
> | avg_gemini-3-pro-preview  | 0.8696  | 0.8742   |
> | vote                      | 0.8692  | 0.8689   |
>
> Based on these results, we select the simple average of all six scores (strategy **avg**  ) as our final aggregation method, since it achieves the highest Pearson and Spearman correlations with human expert scores.
>
> Finally, for each model, we run three independent trials on the full set of 30 proof problems, apply the same scoring method and report the best performance among the three runs. These results appear in the table. Similarly, we normalize the total score of 150 points for the thirty problems to a scale of 100 points:
> | model                         | score   |
> |-------------------------------|-------|
> | gemini-2.5-pro                | 85.45 |
> | o3-high                       | 82.89 |
> | o3                            | 77.11 |
> | deepseek-r1                   | 67.33 |
> | claude-3.7-sonnet-thinking    | 66.67 |
> | doubao-seed-1.6-thinking      | 66.23 |
> | o4-mini-high                  | 63.33 |
> | o3-mini-high                  | 59.33 |
> | claude-4.0-sonnet             | 59.32 |
> | claude-4.0-sonnet-thinking    | 59.00 |
> | doubao-seed-1.6               | 58.23 |
> | claude-3.7-sonnet             | 57.77 |
> | o4-mini                       | 57.67 |
> | o3-mini                       | 56.67 |
> | o1                            | 52.55 |
> | deepseek-v3                   | 44.67 |
> | gpt-4.1                       | 44.43 |
>
>
>
> Since our proof problems are open ended and do not have a single standard solution, fully automated evaluation is indeed challenging. Providing a detailed scoring rubric for each problem and allowing advanced LLMs to grade with reference to these rubrics offers a feasible alternative. This approach helps mitigate the difficulty of automatic evaluation in our benchmark. We will upload all scoring rubrics and problems to the anonymous repository, which you can access at https://anonymous.4open.science/r/aicrypto-iclr-BDE4/ (there may be some delays due to temporary issues with the site). The latest results are included in the revised submission.
>
> Note: Because the gemini-2.5-pro-preview model is no longer available, we use gemini-2.5-pro instead, replacing all its results (all experiments have been rerun).
>
> Thank you again for your thoughtful review. Please let us know if you have any additional questions or concerns.

---

> > ### Author Response · Authors · 2025-11-23
> > **Supplementary Material for Proof Problems**
> >
> > Due to issues with the anonymous.4open.science website, we upload the proof problems to the Supplementary Material, together with reference solutions and scoring rubrics. We warmly welcome you to examine the quality and difficulty of the proof problems.

---

### Meta-Review · Area_Chair_Dkf4 · 2026-01-09

**Summary:**

Reviewers found the benchmark concept timely and appreciated the overall construction pipeline and breadth of evaluated models. However, the decision to reject was driven by concerns about benchmark validity and reliability: fragile/time-sensitive contamination safeguards, limited per-subcategory sample sizes and coverage (raising questions about statistical significance and representativeness), and evaluation methodology that was initially manual and lacked strong reliability reporting. Additional experimental design gaps—e.g., the impact of helper scripts and the agent framework on CTF performance—also reduced confidence that the reported results cleanly reflect model capability rather than evaluation setup.

**Reviewer Concerns:**

The rebuttal substantially improved the paper by correcting the MCQ count, expanding proof problems (18→30), and presenting an automated proof-grading pipeline with rubrics and strong correlation with human scores. These updates address major concerns about dataset size and proof evaluation scalability. Nonetheless, key issues remain: the contamination mitigation strategy remains time-sensitive and not fully future-proof; concerns about limited coverage and small per-category sample sizes are only partially alleviated; and the paper still lacks ablations/sensitivity analyses (helper scripts, agent framework choice) and stronger evidence for grading reliability beyond correlation (e.g., inter-rater agreement and calibration). Overall, while the rebuttal strengthened the submission, it did not fully resolve the core validity concerns underlying the reject decision.

**Reviewer Scores:**

Reviewer jvuN (2): likely unchanged (2), as their primary concerns (contamination robustness, significance/coverage) remain.
Reviewer E18C (4): likely increases slightly (4→5) given the corrected dataset size and improved proof evaluation, though coverage concerns persist.
Reviewer 1oun (6): likely unchanged (6) or slightly lower (6→5) depending on how heavily remaining methodology gaps weigh relative to the improved proof grading.
Reviewer w8pB (4): likely increases modestly (4→5) due to proof expansion and automated scoring, but missing ablations and remaining significance issues would keep them below clear accept.

---

### Decision · Program_Chairs · 2026-01-26

Reject